



# In-situ observation of warm atmospheric layer and the contribution of suspended dusts over the Tarim Basin

Chenglong Zhou[1,2,3], Yuzhi Liu[1,2*], Qingzhe Zhu[1], Qing He [3], Tianliang Zhao[4], Fan Yang[1,2,3], Wen Huo[3], Xinghua Yang[3], Mamtimn Ali[3]

[1]Key Laboratory for Semi-Arid Climate Change of the Ministry of Education, College of Atmospheric Sciences, Lanzhou University, Lanzhou, 730000, China

[2]Collaborative Innovation Center for Western Ecological Safety, Lanzhou University, Lanzhou, 730000, China

[3]Institute of Desert Meteorology, China Meteorological Administration/Taklimakan National Station of Observation and Research for Desert Meteorology in Xinjiang/Taklimakan Desert Meteorology Field Experiment Station, China Meteorological Administration, Urumqi 830002, China

[4]Collaborative Innovation Center on Forecast and Evaluation of Meteorological Disasters, Key Laboratory for Aerosol-Cloud-Precipitation of China Meteorological Administration, Nanjing University of Information Science and Technology, Nanjing 210044, China

*Correspondence to:* Yuzhi Liu (liuyzh@lzu.edu.cn)

**Abstract.** Basing on the radiosonde observations in the spring and summer during 2016-2017, an anomalous warm atmospheric layer is verified and the contribution of suspended dusts over the Tarim Basin (TB) is quantified. The result indicates a warm atmospheric

layer between 300 hPa and 500 hPa with an average intensity of 2.53 K and 1.39 K in the spring and summer, respectively. Over the TB, where the world's second largest moving desert, the Taklimakan Desert (TD) is distributed, large amounts of dust particles are emitted from the TD and suspended over the TB. Using the Cloud-Aerosol Lidar and Infrared Pathfinder Satellite Observations (CALIPSO) data, we found the dusts can be

lifted to the upper atmospheric layer between 2.5 and 5.5 km above mean sea level over the TB. Consequently, the suspended dusts can exert a maximum heating effect of approximately +0.45 K and +0.25 K in spring and summer, respectively. The contribution of dust heating to the anomalous warm atmospheric layer over the TB is 13.77% and 10.25% in spring and summer, respectively. In view of the topographical feature, the TB

is adjacent to the Tibetan Plateau (TP) which acts as an elevated heat source in spring and summer. The warm atmospheric layer over the TB seems a northward extension of Tibet heat source, the concept of which is proposed in this study. Such a northward extension of the elevated heating by the Tibetan Plateau could induce some profound impacts on the regional climate, especially on the western section of the "Silk Road Economic Belt",

and therefore demands more attention.

**Keywords:** Suspended dust, Heat source, Northward extension, Tibetan Plateau, Tarim Basin




# 1  Introduction

The Tarim Basin (TB), which neighbors the Tibetan Plateau (TP) located to the south, covers an area of $5.3 \times 10^5$ km$^2$ and contains the Taklimakan Desert (TD), which occupies the main part of the TB and is one of the major dust sources in Asia (Gong et al., 2003;

Wang et al., 2005). Basing on Cloud-Aerosol Lidar and Infrared Pathfinder Satellite Observations (CALIPSO) data, Liu et al. (2019) pointed out that large amounts of dust particles were emitted from the TD. Moreover, it was found that dust over the TB can be suspended for long time (Huang et al., 2008; Ge et al., 2014; Cheng et al., 2020). Therefore, it is an important issue to investigate the distribution of suspending dusts and

impacts on the Earth-climate system.

Dust is one kind of the absorption aerosols, which absorbs radiation and heats the atmosphere (Yang et al., 2009). It can change the vertical distribution of radiation energy, and then affect the global and regional climate (Huang et al., 2011). The atmospheric layer is heated where the dust is suspended (Huang et al., 2006a, 2006b; Huang et al., 2015;

Liu et al., 2014; Jia et al., 2018). The radiative forcing of dust over the TB and its feedback on climate change in Central Asia are a series of scientific issues that need to be further clarified. Hence, quantifying the heating induced by dust in the upper atmosphere over the TB is necessary.

About the radiative effect of suspending dusts over the TB, previous studies mainly

based on the satellite observation and numerical model. Using CALIPSO data, Huang et al. (2009) found the significant radiative forcing and heating effect due to dusts over the part of the TB. Gu et al. (2006, 2016) and Law et al. (2006) present some numerical results to elucidate the dust's impact on air temperature at upper layers. However, focusing on above topic, the evidence from in-situ observations is lack.


In this study, ground-based radiosonde observations, reanalysis and satellite data in spring and summer during 2016-2017 are employed to investigate the vertical distribution of the air temperature and the three-dimensional structure of suspended dust over the TB. Furthermore, the heating effect of suspended dust on the atmosphere is quantified. At last, the concept of the northward extension of Tibetan heat source is proposed.

The rest of the paper is organized as follows: section 2 describes the data and methodology used in the study. Section 3 presents the main results and discussion. Conclusions are given in section 4.

## 2    Data and Methodology

### 2.1    Radiosonde data

In this study, radiosonde data obtained from 6 radiosonde stations in the Tarim Basin (TB) (Fig. 1), including Kashi (KS), Akesu (AKS), Kuerle (KEL), Ruoqiang (RQ), Minfeng (MF) and Hetian (HT), acquired in spring and summer during 2016-2017 are used to obtain temperature profiles. During the radiosonde observations, temperature profiles were measured twice a day (08:00 and 20:00 UTC+8). The temporal resolution

of the radiosonde data is 1 minute. Then we process the radiosonde data into monthly averaged at intervals of 25 hPa from 850 hPa to 200 hPa through the calculation. Simultaneously, weather phenomenon records from the ground-based meteorological stations are used for classification and statistics of the dusty weather all 6 stations (KS, AKS, KEL, RQ, MF and HT).

### 2.2    Reanalysis data


The Fifth-Generation European Centre for Medium-Range Weather Forecasts (ECMWF) Reanalysis (ERA-5) and Modern-Era Retrospective Analysis for Research and Applications, Version 2 (MERRA-2) reanalysis data in spring and summer during 2016-

2017 are used. The ERA-5 reanalysis data in this study have a 0.5°×0.5° latitude/longitude

spatial resolution and 37 pressure levels in the vertical direction. The MERRA-2 data are

collected on a regular latitude-by-longitude grid of 0.5°×0.625° with 42 pressure levels

in the vertical direction. The temporal resolutions of the temperature from the two sets of

reanalyzes are 1 month. The ERA-5 data are the latest global atmospheric reanalysis

produced by the ECMWF based on the Integrated Forecasting System (IFS) Cy41r2

(Hersbach et al., 2020). Notably, the ERA-5 data are generated from an ECMWF IFS

spectral model and do not yet assimilate the impact of aerosols on meteorology (Simmons,

2006). The MERRA-2 data are the update of NASA's previous satellite reanalysis system

and include additional observations and improvements to the Goddard Earth Observing

System, Version 5 (GEOS-5) Earth system model (Randles et al., 2017). Unlike the ERA-

5 data, the MERRA-2 data include the impact of dusts on meteorology (Gelaro et al.,

2017).

### 2.3 Satellite data

Cloud-Aerosol Lidar and Infrared Pathfinder Satellite Observations (CALIPSO) was

launched on April 28, 2006 to study the impact of clouds and aerosols on the Earth's

radiation budget and climate. The CALIPSO satellite comprises three instruments, the

Cloud-Aerosol Lidar with Orthogonal Polarization (CALIOP), the Imaging Infrared

Radiometer (IIR), and the Wide Field Camera (WFC). In this study, the CALIPSO Level

1B and Level 2 Vertical Feature Mask (VFM) data sets (aerosol profile), which contain a

half-orbit (day or night) of calibrated and geolocated single shot (highest resolution) Lidar

profiles, were used to detect dust events. The CALIPSO Level 1B product provides the

profiles of the total attenuated backscatter at 532 and 1064 nm; the feature classification

from CALIPSO Level 2 VFM was used to distinguish the types of aerosols. Meanwhile,



the seasonally averaged CALIPSO Level 2 VFM product was used to identify the dust

profile top height.

**2.4   Method of distinguishing the anomalous warm atmospheric layer**

First, based on the radiosonde temperature data from each observation station in the

TB in spring and summer during 2016-2017, the temperature can be fitted by the

following equation:

$$T_F = aH + b \qquad (1)$$

Where, $T_F$ is the fitting temperature (℃), $a$ is the mean slope, $H$ is the altitude (hPa),

and $b$ is a constant. Then, according to Eq. (1), the fitting temperature profile can be

calculated from the altitude data. Table 1 shows the fitting equations of each station.

The temperature difference ($\Delta T$) is calculated by comparing the radiosonde

measurements with the temperature computed from the fitting equation.

$$\Delta T = T_O - T_F \qquad (2)$$

Where, $T_O$ is the radiosonde temperature, $T_F$ is the fitting temperature based on the

Eq 1. This method serves as a good indicator of the anomalous temperature variation. A

positive value means that the atmosphere is heating, while a negative value means that

the atmosphere is cooling. The absolute value represents the intensity.

2.5. Method of estimating the dust effect on the temperature

In the analysis, the dust effect on the temperature is estimated based on the

observation minus reanalysis (OMR) method proposed by Ding et al. (2013, 2016).

$$OMR = T_O - T_R \qquad (3)$$

Where, $T_O$ is also the radiosonde temperature, $T_R$ is the ERA-5 temperature, which

does not include the impact of dust aerosols and assimilates only limited upper

atmospheric measurement data (Simmons, 2006). Therefore, this method serves as a good



indicator of the dust heating. A positive value means that the atmosphere is heating, while a negative value means that the atmosphere is cooling. The absolute value represents the intensity.

## 3 Results and discussions

### 3.1 Anomalous warm layer over the TB

In the troposphere, the atmospheric temperature (hereinafter referred to simply as temperature) generally decreases linearly with increasing altitude. However, the temperature shows an anomalous temperature lapse rate over the TB. Figs. 2a-2f present

profiles of the temperature difference ($\Delta T$) between the radiosonde observations and the mean temperature rate calculations (see Section 2) at the observation stations, namely, Kashi (KS), Akesu (AKS), Kuerle (KEL), Hetian (HT), Minfeng (MF), and Ruoqiang (RQ), located in the TB in spring and summer during 2016-2017. Positive and negative values indicate warming and cooling, respectively. A relatively warm layer is observed

between 700 hPa and 300 hPa in the spring and summer over the TB (shading in Figs. 2a-2f). However, the temperature difference presents obvious discrepancies among the stations. According to the location of 6 observational stations, as shown in Fig. 1, we divide the TB into north (KS, AKS and KEL) and south (HT, MF and RQ) region. In the north of the TB, although the height of the warm layer is consistent in both seasons, the

warming is more intense in spring than in summer. Comparatively, in the south of the TB, the warming phenomenon extends to higher altitudes in summer than in spring, with greater warming at an altitude of more than 400 hPa.

In this study, we focus mainly on the temperature anomalies at altitudes of 500-300 hPa. The mean values of $\Delta T$ between 500 hPa and 300 hPa at each station in spring and

summer are shown in Figs. 3a and b, respectively. In spring, the values of $\Delta T$ at each

station are positive and range from 2.29 K to 2.73 K, with a mean value of 2.53 K. In

comparison, $\Delta T$ is smaller in summer than in spring, varying from 1.21 K to 1.57 K, with

a mean value of 1.39 K. Overall, an anomalous warm layer is measured at altitudes

between 500 hPa and 300 hPa over the TB.

**3.2 Distribution of dust aerosols in the warm layer over the TB**

Dust aerosols are the principal particulate type in the atmosphere over the TB (Cheng

et al., 2020). The distribution of dust aerosol is the key factor to evaluate its radiation

forcing. In the following, the vertical distribution and variation of dust are analyzed based

on the Level 1B and Level 2 Vertical Feature Mask (VFM) data sets over the TB from

Cloud-Aerosol Lidar with Orthogonal Polarization (CALIOP). The distribution of the

dust profile top height over the TB is also presented.

The product of CALIPSO, which can observe aerosols over bright surfaces and

beneath thin clouds in clear skies (Vaughan et al., 2004; Winker et al., 2006), was used to

identify dust aerosols. With a total attenuated backscatter coefficient at 532 nm and

classified particles from CALIPSO were combined to identify dust aerosols. To identify

dust aerosols, values of 0.0008-0.048 $km^{-1}$ $sr^{-1}$ was chosen as the thresholds of the total

attenuated backscatter (Liu et al., 2015; Jia et al., 2015). Fig. 4 (left panels) shows the

CALIPSO orbit-altitude cross-section of the 532-nm total attenuated backscattering

coefficient on 4 July, 5 July, 25 July and 27 July 2016 along the CALIPSO trajectory

presented in Fig. 1. The gray shading in Fig. 4 indicates the topography, and the deep blue

area denotes the absence of signal due to clouds, which the laser cannot penetrate. As

shown in Fig. 4 (left panels), the total attenuated backscatter ranged from 0.002 to 0.005

$km^{-1}$ $sr^{-1}$. Based on the thresholds for identifying dust aerosols, 4 July and 5 July 2016 are

considered two severe dusty days. Meanwhile, the dust layer can also be distinctly





identified during the clear day, however, the range and intensity is relatively smaller

compared with the dusty day.

CALIPSO data reveals that vertically extended dust layers are widespread

throughout the TB with peak lidar returns between 2.5 and 5.5 km above mean sea level

(MSL) due to the strong convective activity during the dusty and clear day (Cheng et al.,

2020). The results were consistent with other studies (Huang et al., 2009; Liu et al., 2015).

Fig. 4 (right panels) also describes the thick dust plumes are observed over the entire TB.

Moreover, the geographic setting of the basin, surrounded by high mountains, generates

atmospheric circulations in the basin that are favorable for dust to remain suspended in

the air for a long time (Tsunematsu et al., 2005).

Fig. 5a plots the frequencies of dust events observed by ground stations in spring

and summer during 2016-2017 throughout the TB. In spring, the frequencies of dust

events are 55.43%, 50.00%, 53.26%, 33.70%, 22.83% and 34.24% in MF, HT, RQ, KS,

KEL and AKS respectively. In summer, the frequencies are 60.33%, 42.93%, 50.54%,

2.72%, 4.89% and 17.39% in MF, HT, RQ, KS, KEL and AKS respectively. Therefore,

the frequencies of dust events south of the TB are obviously higher than those north of

the TB. Here, dust events include dust storms and cases of blowing and floating dust, of

these dust events, cases of floating dust occupy the majority, accounting for more than

50.00% in the south of the TB (Fig. 5b). The results above are consistent with the findings

of Zhou et al. (2020), and the main reason is when the cold air streams of different

intensities intrude, the wind fields converge strongly and rise in the HT and MF areas,

thereby making these areas with the highest frequency of dust weather (Han et al., 2005).

The information on the dust top height (DTH) can help elucidate the vertical

structure of dust. The DTH, which is defined as the height above surface elevation (a.s.e),


shows significant seasonal variations over TB (Figs. 5c and 5d). The blank area is the

default value of observation. Over the TB, the DTH was larger in spring compared to the

summer, with the range of 2.0-5.5 km. Note that the variations here resembled those of

the boundary layer height (BLH) (Luo et al., 2017). Previous study suggested that BLH

was a key factor to determine the vertical distributions of dusts in the TB. Different from

urban district (Ding et al. 2016; Huang et al. 2018; Li et al. 2021), the BLH was super

high with over 5,000-m-depth in the TD based on the sounding data obtained from a

month-long intensive field campaign carried out in July 2016 (Wang et al., 2019).

The results above indicate that spring and summer are the frequent seasons of dust

weather in TB, especially in the south of the basin. Dust can suspend in the upper layer

for a long time. Moreover, as an important component of absorbing aerosols, the dust

aerosols could provide an elevated heat source to the air (Lau et al., 2006). Therefore, we

reveal the contribution of the suspended dust to the anomalously warm layer over the TB

based on the in-situ observation in the following.

### 3.3 Spatial and temporal features of the heating effect due to dust aerosols

Previous studies reported that the dust emitted from the TD has anomalously strong

optical absorption properties and thus a more significant heating ability (Ge et al., 2010;

Huang et al., 2015). In the following, based on the method proposed by Ding et al. (2013,

2016) using the temperature difference between the observation and reanalysis data

(referred to as the OMR: observation minus reanalysis, see the section 2 Data and

Methodology), the effects of dust aerosols on the temperature are estimated. Here,

radiosonde observations and Fifth-Generation European Centre for Medium-Range

Weather Forecasts (ECMWF) Reanalysis (ERA-5) reanalysis data are used.

In this study, we focus mainly on the temperature anomalies at altitudes of 500-300

hPa. Fig.6 present vertical profiles of the temperature difference between the radiosonde observations and the ERA-5 data at altitudes of 500-300 hPa in spring and summer during

2016-2017. The ERA-5 reanalysis does not assimilate aerosol data (Ding et al., 2013, 2016), while radiosonde measurements include the influence of aerosols. Thus, the effect of aerosols on the temperature can be estimated by calculating the OMR. As shown in Fig. 6, dust can heat the layer at altitudes of 500-300 hPa in spring and summer over the TB, however, it shows obvious temporal and spatial variation characteristics.

In spring, all stations show the heating from 500 hPa to 300 hPa with the mean OMR of 0.30 K. In summer, the initial height of the heating layer is higher than that in spring at altitudes of 500-300 hPa, especially in HT and MF. The average OMRs present that the heating layer is between 400 hPa and 300 hPa with the mean intensity of 0.13 K. The results indicate that the heating intensity in summer is significantly weaker than that

in spring.

In the south of TB, the average OMR of the heating layer in spring and summer are 0.31 K and 0.17 K, respectively. In the north of TB, the average OMR of the heating layer in spring and summer are 0.28 K and 0.12 K, respectively. Therefore, the heating intensity in the south is stronger than that in the north over the TB.

Although the heating effect of suspend dust is confirmed from the in-situ observation of the station, the dust heating of the whole basin cannot be obtained due to the limitation of the number of stations. Therefore, in order to solve this problem, we take the Modern-Era Retrospective Analysis for Research and Applications, Version 2 (MERRA-2) reanalysis data as the observation. The main reasons are as follows: firstly, one of the

advances in MERRA-2 is the assimilation of aerosol observations, thereby it provides a multidecadal reanalysis in which aerosol and meteorological observations are jointly



assimilated within a global data assimilation system (Gelaro et al., 2017). Secondly, Fig. 7a shows the relationship between atmospheric radiosonde temperature observations and MERRA-2 reanalysis data at altitudes of 500-300 hPa in spring and summer during 2016-

2017. The linear fitting slopes between Atmospheric radiosonde temperature observations and MERRA-2 reanalysis data are 0.993 and 0.995 in 2016 and 2017, respectively, with the correlation coefficients ($R^2$) of 0.997. In addition, the mean square errors (MSE) of each layer between the radiosonde observation and MERRA-2 data at altitudes of 500-300 hPa in 2016-17 are obtained (Fig. 7b). The MSE is used to measure the deviation of

two sets of data. It is found that the MSE of each layer is between 0.06 and 0.10. Therefore, it is feasible that we use MERRA2 data as observations to analyze the heating effect of suspend dust over the whole basin.

Figs. 7c and 7d show the distributions of OMR averaged over 500-300 hPa in spring and summer during 2016-2017. The mean OMRs at altitudes between 500 hPa and 300

hPa present relatively high values south of the TD in both spring and summer, which are basically consistent with the dust occurrence frequency (Figs. 5a and 5b). Accordingly, the heating intensity by dust is greater (with a maximum of approximately +0.45 K) in spring than in summer (with a maximum of approximately +0.25 K). In summary, dust aerosols can exert a heating effect and are attributable to the warming in the atmospheric

layer between 500 hPa and 300 hPa over almost the entire TB. As illustrated above, the results clearly demonstrate that dust can heat the upper atmosphere over the TB. Suspended dust serves as a critical "bridge". This is a unique atmospheric phenomenon in China.

Combined with Figs. 2 and 6, Table 2 shows the contribution of suspend dust to

abnormal heating layer at altitudes of 500-300 hPa over TB, with the average contribution


of 13.77% and 10.25% in spring and summer, respectively. It is obvious that the dust radiative forcing is one of the contributors for the warm layer over the TB. We find that the water vapor at the altitude of 4-7 km is nearly saturated at daytime over the TZ (Fig omitted), which is almost at the same height as the dust layer, the water vapor would

absorb solar radiation and heat the atmosphere. Hence, the water vapor may be an important contributor for the warm layer over the TB. Furthermore, radiative properties of the surface, atmospheric trace gases and clouds also influence the aerosol-radiation interactions (IPCC, 2013). Therefore, the warm layer over the TB is the result of multiple factors. There are still many unknown issues, which are worthy to be further studied.

**3.4 Concept of the northward extension of Tibetan heat source**

Topographically, the TB is adjacent to the TP (Fig. 1), which acts as an elevated heat source in spring and summer (Duan et al., 2013; Wonsick et al., 2014). The warm atmospheric layer over the TB seems a northward extension of Tibetan heat source. Therefore, we put forward the concept of the Tibetan heat source's northward extension,

which is illustrated in Fig. 8. Considering the important roles of the TP and TB in affecting the climate along the Silk Road Economic Belt (Liu et al., 2020; Zhao et al., 2020), more attention should be paid to the impact of the Tibetan heat source's northward extension on the regional climate. Moreover, the northward extension of the Tibetan heat source could be attributed to extreme weather. In recent years, heavy precipitation events have

occurred frequently in Xinjiang, ie., the extreme precipitation event lasted for more than 100 h in KS, AKS and HT from May 15 to 21, 2018. During this extreme precipitation event, the precipitation broke the annual historical extremum in many places, seriously endangering the local economic development and people's lives. Northward extension of the Tibetan heat source and its thermal forcing effect on regional precipitation anomaly

in spring and summer need further analysis, meanwhile, it is also a new issue to be studied.

Therefore, an in-depth study of the influence of the Tibetan heat source's northward

extension on the regional weather and climate needs to be performed in the future.

## 4  Conclusions

Dust aerosols can warm the climate (Figs. 6 and 7), but how much dust aerosols net

influence global climate is highly uncertain (Penner, 2019). Aerosol-radiation interactions

requires knowledge of the spectrally varying aerosol extinction coefficient, single

scattering albedo, and phase function (McComiskey and Feingold, 2008; Loeb and Su,

2010; Kahn, 2012), which can in principle be estimated from the aerosol size distribution,

shape, chemical composition and mixing state (Sicard et al., 2014; Lacagnina et al., 2015;

Li and Sokolik, 2018). They lead to the large uncertainties in quantifying the dust

radiative effect in the models. Different from other studies (Gu et al., 2006, 2016; Law et

al., 2006; Huang et al., 2009), the in-situ observational evidences on dust aerosols' heating

effect over TB for the first time in this study. Of course, although we have avoided these

complex processes above, it is undeniable that errors still exist. The main conclusions are

as follows:

A relatively warm layer is observed between 700 hPa and 300 hPa in the spring and

summer over the TB. In this study, we focus on the temperature anomalies at altitudes of

500-300 hPa, the values of $\Delta T$ at each station are positive and range from 2.29 K to 2.73

K in spring. In comparison, $\Delta T$ is smaller in summer than in spring, varying from 1.21 K

to 1.57 K.

Dust can heat the layer at altitudes of 500-300 hPa in spring and summer over the

TB, which shows obvious temporal and spatial variation characteristics. The heating

intensity in summer (with the mean OMR of 0.30 K) is significantly weaker than that in

spring (with the mean OMR of 0.13 K). Over the south of TB, the average OMR of the

heating layer in spring and summer are 0.31 K and 0.17 K, respectively. Comparatively,

over the north of the TB, the average OMR of the heating layer in spring and summer are

0.28 K and 0.12 K, respectively. Dust radiative forcing is one of the contributors for the

warm layer over the TB. Topographically, the TB is adjacent to the Tibetan Plateau (TP),

which acts as an elevated heat source in spring and summer. The warm atmospheric layer

over the TB seems a northward extension of Tibet heat source. Therefore, the concept of

the northward extension of Tibetan heat source is proposed.

**Code availability.**

The data and data analysis method are available upon request.

**Data Availability.**

The ERA-5 reanalysis data were available at ECMWF (https://cds.climate.copernicus.eu

/cdsapp#!/dataset/reanalysis-era5-pressure-levels-monthly-means?tab=form) and MERR

A-2 reanalysis data were provided by NASA Goddard Earth Science Data and Informati

on Services Center through the NASA GES DISC online archive (air temperature, https:

//goldsmr5.gesdisc.eosdis.nasa.gov/data/MERRA2_MONTHLY/M2IMNPASM.5.12.4/;

dust mixing ratio, https://goldsmr5.gesdisc.eosdis.nasa.gov/data/MERRA2/M2I3NVAE

R.5.12.4/). The CALIPSO data were obtained from the National Aeronautics and Space

Administration (NASA) Langley Research Center Atmospheric Sciences Data Center (h

ttps://www-calipso.larc.nasa.gov/products/lidar/browse_images/production/). The meteo

rological observation data were supplied by the National Meteorological Information Ce

nter (http://data.cma.cn/) under license and so cannot be made freely available.

**Supplement.**

The supplement related to this article is available online at:

**Author contributions.**

Yu-zhi Liu designed the study and contributed ideas. Qing-zhe Zhu, Qing He and Fan Yang conducted the long-term measurements and provided the data. Cheng-long Zhou, Yu-zhi Liu and Tian-liang Zhao interpreted the data. Cheng-long Zhou contributed to the interpretation and writing of the manuscript with contributions from the coauthors.

**Competing interests.**

The authors declare no competing financial interests.

**Acknowledgements.**

The authors are grateful to the science teams for providing the accessible data products used in this study.

**Financial support.**

This work was supported by the National Natural Science Foundation of China (42030612) and jointly supported by the Strategic Priority Research Program of the Chinese Academy of Sciences (Grant No. XDA2006010301), National Natural Science Foundation of China (41905009, 91744311, 41991231, 91937302, 41875019, 41975010 and 41830968), and the Fundamental Research Funds for the Central Universities (lzujbky-2020-kb02).

Review statement.

This paper was edited by and reviewed by referees.






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



**Figure captions**

Figure 1.    Topographical map of the study domain and distribution of observation stations in TB; the contours of the terrain height are in km (above mean sea level). The solid lines indicate the trajectory of the CALIPSO satellite over the TB on 4 July (20:22 UTC, black line), 5 July (07:02 UTC, red line), 25 July (20:40 UTC, blue line,) and 27 July 2016 (20:28 UTC, green line), of which, 4 July and 5 July 2016 are dusty weather, 25 July and 27 July 2016 are clear day.

Figure 2.   Temperature difference ($\Delta T$) profiles in spring and summer during 2016-2017 at the (a) Kashi (KS), (b) Akesu (AKS), (c) Kuerle (KEL), (d) Hetian (HT), (e) Minfeng (MF), and (f) Ruoqiang (RQ) observation stations. The shading in (a-f) indicates the layer with positive $\Delta T$.

Figure 3.    Averaged $\Delta T$ between 500 and 300 hPa in (a) spring and (b) summer. The position of the circle represents the site location, and the size of the circle represents the heating intensity.

Figure 4.    The altitude-orbit cross-section of 532 nm total attenuated backscattering intensity (left panels) and classified particles (right panels) on (a1 and b1) 4 July, (a2 and b2) 5 July, (a3 and b3) 25 July and (a4 and b4) 27 July 2016 along the trajectory of the CALIPSO satellite over the TP, as presented in Figure 1. The gray shading indicates the topography.

Figure 5.   (a) Frequencies of dust events at the observation stations (KS, AKS, KEL, HT, MF and RQ); (b) Frequencies of dust events including dust storms and cases of blowing and floating dust at the observation stations (KS, AKS, KEL, HT, MF and RQ) in spring and summer during 2016-2017. The blue dashed line represents the boundary between spring and summer; (c) Seasonal distribution





of dust Top Height (km) obtained from CALIPSO data in spring over the TB; (d)

580   Same as (c) but for summer.

Figure 6. Profiles of the temperature difference (radiosonde observation minus ERA-5 data) in spring and summer during 2016-2017 at (a) Kashi (KS), (b) Akesu (AKS), (c) Kuerle (KEL), (d) Hetian (HT), (e) Minfeng (MF) and (f) Ruoqiang (RQ).

585 Figure 7. (a) Relationship between atmospheric radiosonde temperature observations and MERRA-2 reanalysis data at altitudes of 500-300 hPa in spring and summer during 2016-2017; (b) the mean square errors (MSE) of each layer between the radiosonde observation and MERRA-2 data at altitudes of 500-300 hPa; and distributions of the temperature difference between MERRA-2 and ERA-5 data (OMR) averaged over 500-300 hPa in (c) spring and (d) summer during 2016-

590   2017. The black solid circle represents the site location.

Figure 8. Conceptual scheme of the Tibetan heat source's northward extension partially attributed to the heating effect of suspended dust aerosols over the TB. Black lines denote the atmospheric temperature profile, in which the solid and dotted lines indicate the dust-influenced and dust-free profiles, respectively. Yellow arrows

595   denote solar radiation, including the parts reflected and absorbed by clouds, dust aerosols and the surface. Red arrows at the surface denote the sensible heat. White solid arrows show the turbulence and convective mixing in the planetary boundary layer. White dashed lines show the altitude.






Table 1. Fitting equations of each station based on the Eq. (1).

| Station | Season | Fitting equation | |
| --- | --- | --- | --- |
| KS | Spring | $T_F = 0.117H - 77.564$ | $R^2 = 0.983$ |
| | Summer | $T_F = 0.109H - 64.468$ | $R^2 = 0.994$ |
| AKS | Spring | $T_F = 0.115H - 77.786$ | $R^2 = 0.981$ |
| | Summer | $T_F = 0.109H - 65.186$ | $R^2 = 0.992$ |
| KEL | Spring | $T_F = 0.113H - 77.682$ | $R^2 = 0.981$ |
| | Summer | $T_F = 0.110H - 65.349$ | $R^2 = 0.992$ |
| HT | Spring | $T_F = 0.117H - 76.374$ | $R^2 = 0.981$ |
| | Summer | $T_F = 0.106H - 60.881$ | $R^2 = 0.988$ |
| MF | Spring | $T_F = 0.117H - 76.382$ | $R^2 = 0.981$ |
| | Summer | $T_F = 0.106H - 60.502$ | $R^2 = 0.986$ |
| RQ | Spring | $T_F = 0.115H - 76.971$ | $R^2 = 0.981$ |
| | Summer | $T_F = 0.108H - 62.300$ | $R^2 = 0.989$ |







Table 2. Contribution of suspend dust to abnormal heating layer at altitudes of 500-300

615                                                              hPa over TB.

| Station | KS | | AKS | | KEL | |
|---|---|---|---|---|---|---|
| | Spring | Summer | Spring | Summer | Spring | Summer |
| Contribution | 17.51% | 29.00% | 11.83% | 7.87% | 10.30% | 3.16% |
| Station | HT | | MF | | RQ | |
| | Spring | Summer | Spring | Summer | Spring | Summer |
| Contribution | 14.65% | 4.06% | 15.13% | 6.94% | 13.21% | 10.46% |








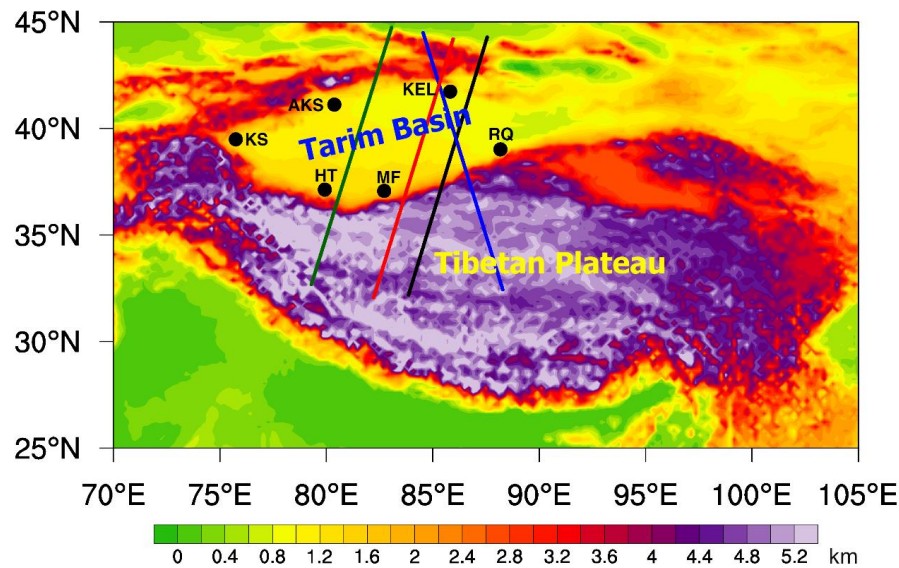

**Figure 1.** Topographical map of the study domain and distribution of observation stations in TB; the contours of the terrain height are in km (above mean sea level). The solid lines indicate the trajectory of the CALIPSO satellite over the TB on 4 July (20:22 UTC, black line), 5 July (07:02 UTC, red line), 25 July (20:40 UTC, blue line,) and 27 July 2016 (20:28 UTC, green line), of which, 4 July and 5 July 2016 are dusty weather, 25 July and 27 July 2016 are clear day.





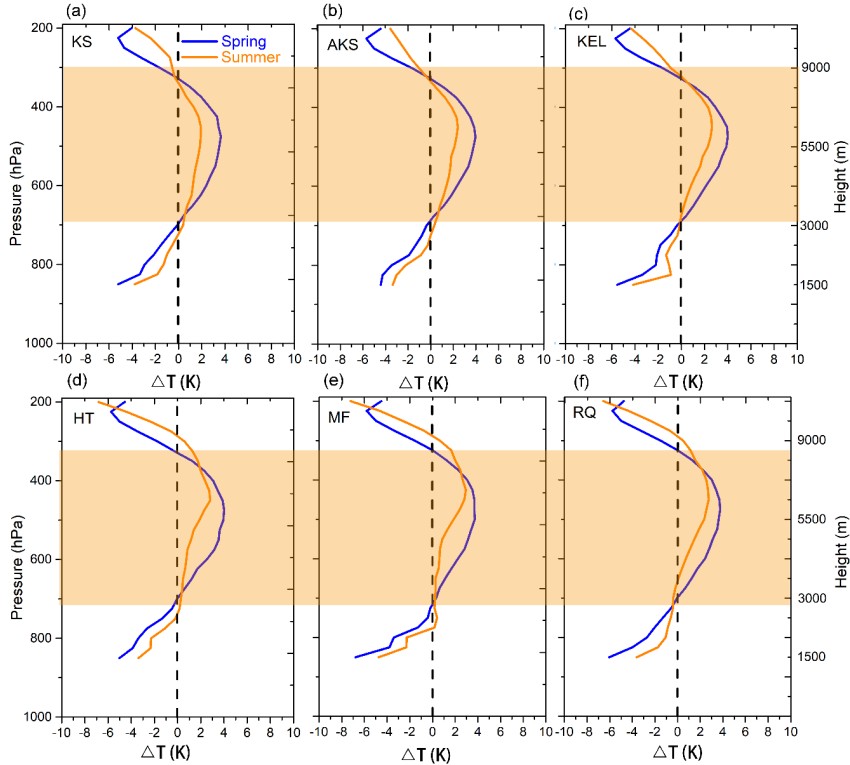

**Figure 2.** Temperature difference (Δ*T*) profiles in spring and summer during 2016-2017 at the (a) Kashi (KS), (b) Akesu (AKS), (c) Kuerle (KEL), (d) Hetian (HT), (e) Minfeng (MF), and (f) Ruoqiang (RQ) observation stations. The shading in (a-f) indicates the layer

with positive Δ*T*.


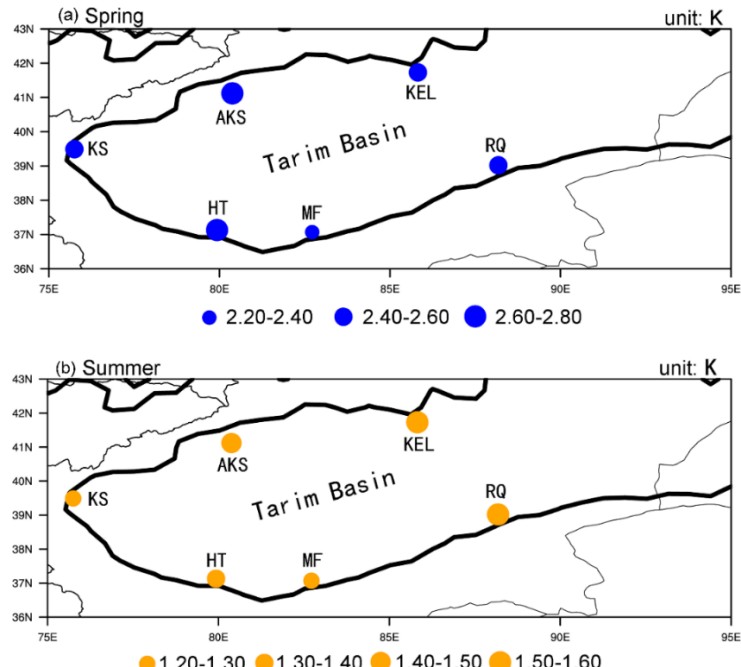

**Figure 3**. Averaged $\Delta T$ between 500 and 300 hPa in (a) spring and (b) summer. The

position of the circle represents the site location, and the size of the circle represents the

heating intensity.







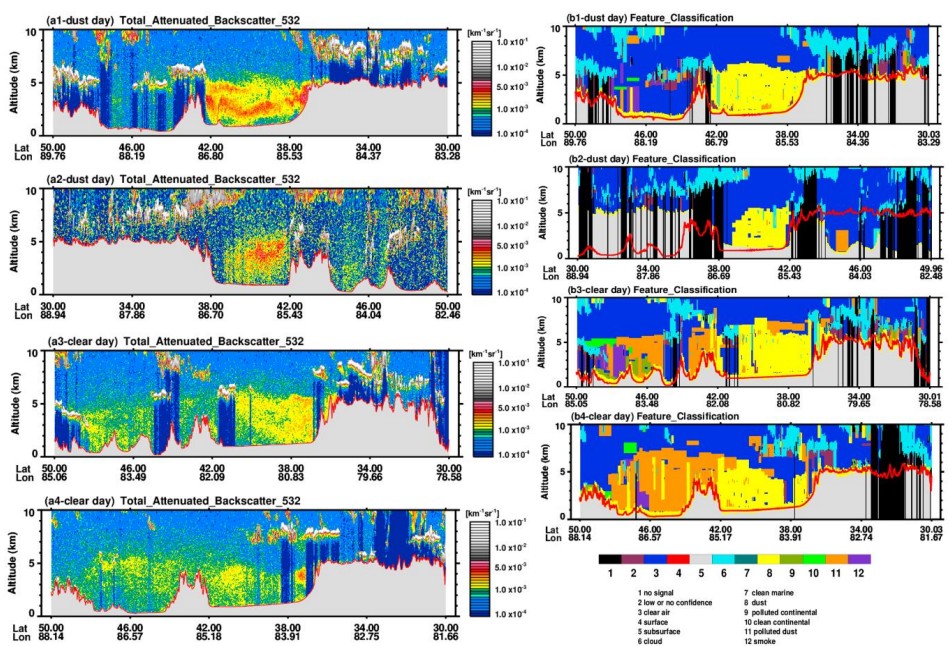

**Figure 4.** The altitude-orbit cross-section of 532 nm total attenuated backscattering intensity (left panels) and classified particles (right panels) on (a1 and b1) 4 July, (a2 and b2) 5 July, (a3 and b3) 25 July and (a4 and b4) 27 July 2016 along the trajectory of the

CALIPSO satellite over the TP, as presented in Fig. 1. The gray shading indicates the topography.


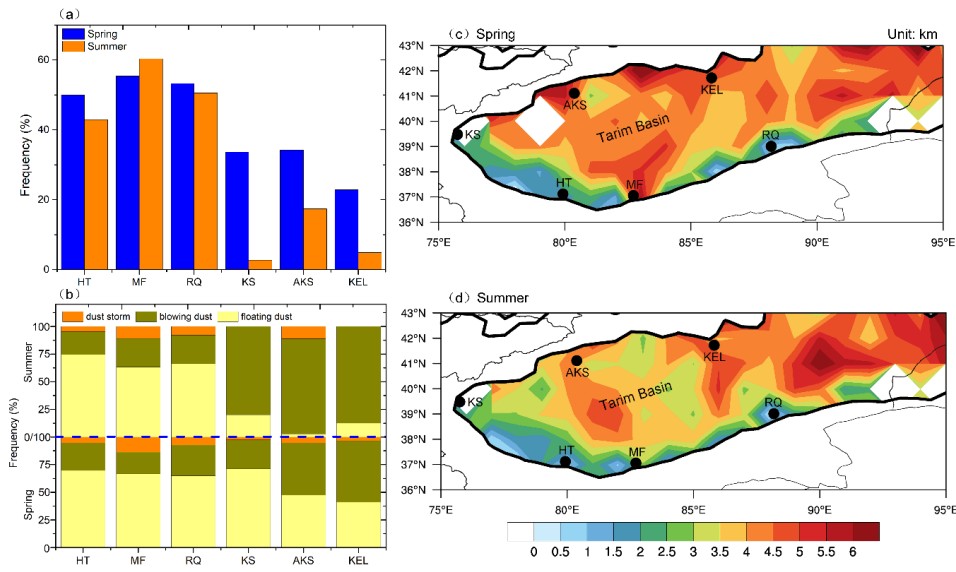


**Figure 5.** (a) Frequencies of dust events at the observation stations (KS, AKS, KEL, HT, MF and RQ); (b) Frequencies of dust events including dust storms and cases of blowing and floating dust at the observation stations (KS, AKS, KEL, HT, MF and RQ) in spring and summer during 2016-2017. The blue dashed line represents the boundary between spring and summer; (c) Seasonal distribution of dust Top Height (km) obtained from CALIPSO data in spring over the TB; (d) Same as (c) but for summer.



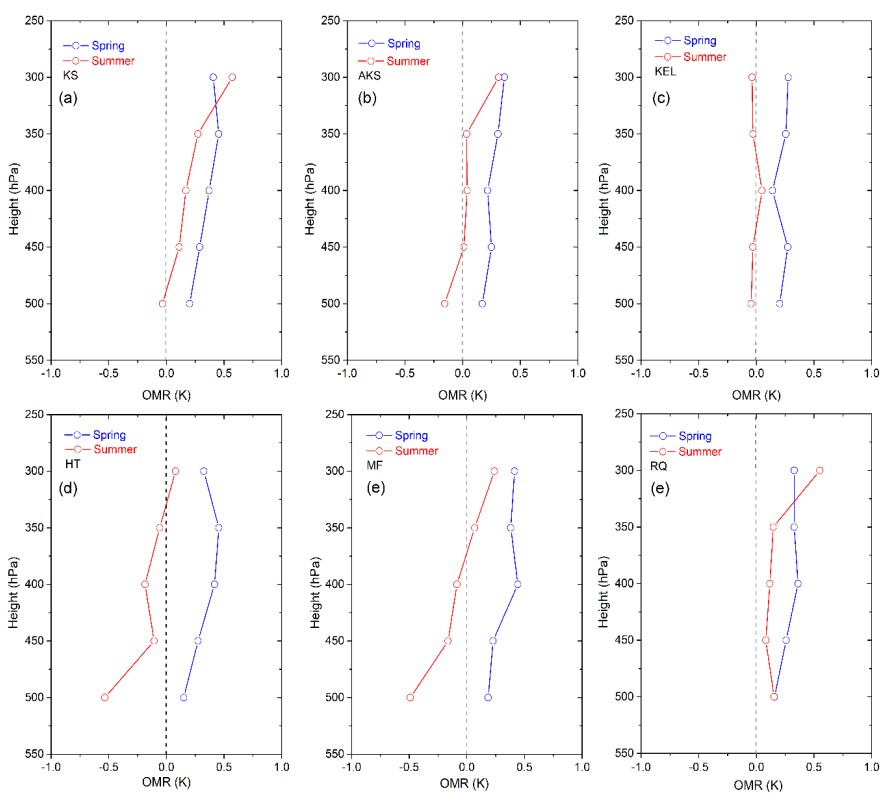


**Figure 6.** Profiles of the temperature difference (radiosonde observation minus ERA-5 data) in spring and summer during 2016-2017 at (a) Kashi (KS), (b) Akesu (AKS), (c) Kuerle (KEL), (d) Hetian (HT), (e) Minfeng (MF) and (f) Ruoqiang (RQ).



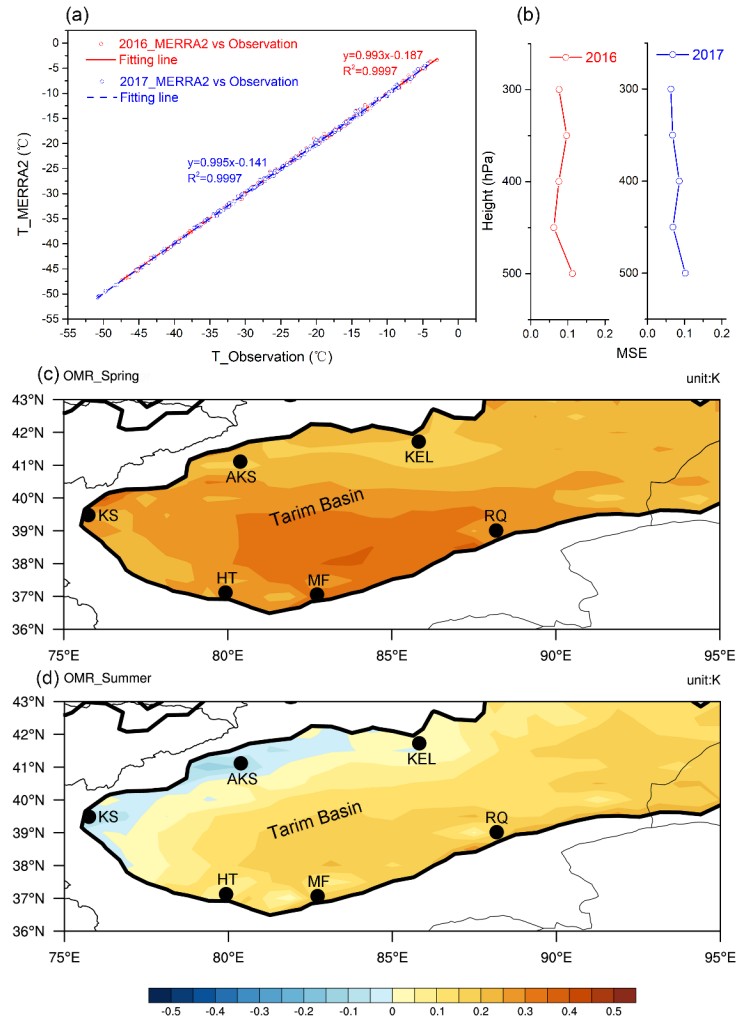

**Figure 7.** (a) Relationship between atmospheric radiosonde temperature observations and MERRA-2 reanalysis data at altitudes of 500-300 hPa in spring and summer during 2016-2017; (b) the mean square errors (MSE) of each layer between the radiosonde observation and MERRA-2 data at altitudes of 500-300 hPa; and distributions of the temperature difference between MERRA-2 and ERA-5 data (OMR) averaged over 500-300 hPa in (c) spring and (d) summer during 2016-2017. The black solid circle represents the site location.


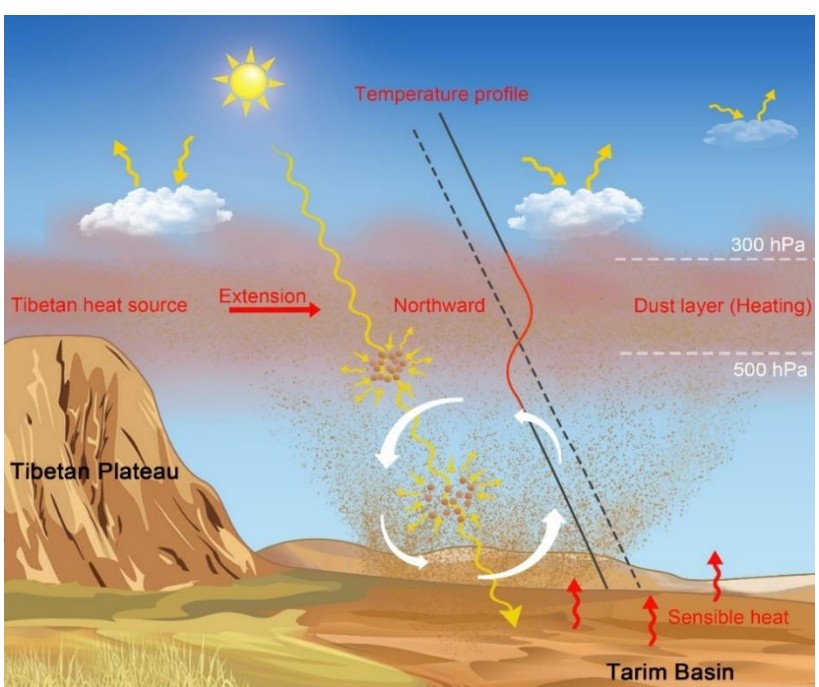

**Figure 8.** Conceptual scheme of the Tibetan heat source's northward extension partially attributed to the heating effect of suspended dust aerosols over the TB. Black lines denote the atmospheric temperature profile, in which the solid and dotted lines indicate the dust-influenced and dust-free profiles, respectively. Yellow arrows denote solar radiation, including the parts reflected and absorbed by clouds, dust aerosols and the surface. Red arrows at the surface denote the sensible heat. White solid arrows show the turbulence and convective mixing in the planetary boundary layer. White dashed lines show the altitude.