# Peer review of "In-situ observation of warm atmospheric layer and the heat contribution of suspended dust over the Tarim Basin"

_Atmospheric Chemistry and Physics, 2021_

## Author Comment (AC1)

Journal: Atmospheric Chemistry and Physics

Article title: In-situ observation of warm atmospheric layer and the heat contribution of suspended dust over the Tarim BasinManuscript number: acp-2021-892

Dear Editors and Reviewers:

Thank you for your letter and for the reviewers' comments concerning our manuscript entitled "In-situ observation of warm atmospheric layer and the heat contribution of suspended dust over the Tarim Basin (acp-2021-892)". These comments are all valuable and very helpful for revising and improving our paper, as well as the important guiding significance to our researches. We have studied comments carefully and have made correction which we hope meet with approval. Revised portion are marked in the document named "manuscript\_revised". The point-to-point responses to reviewer #1's comments are as follows.

**Reviewer #1:**

Comment to "In-situ observation of warm atmospheric layer and the contribution of suspended dusts over the Tarim Basin"

The large amount of dust aerosols from Taklimakan Desert could introduce serious impacts on the atmospheric thermal structure by absorbing solar radiation and then change regional weather and circulation. Using in-situ radiosonde observations along with the CALIPSO satellite observations, this study provides a comprehensive analysis about the warm atmospheric layer observed and examines the contribution form suspended dusts. This is particularly valuable for the aerosol science community and I would like recommend its acceptance for publication after some necessary minor revisions.

**Response:** Thank you so much for your kindness and your encouragement.

1. Line 32, adding "and" would be helpful before "large amounts of"

**Response**: Thanks for your comments. We have modified the sentence as "The Taklimakan Desert (TD), i.e., the world's second largest moving desert, is contained in the TB, emits large amounts of dust particles, which remain suspended over the TB." (Lines 27-29).

**2.** Line 52, I would suggest changing "which neighbors the Tibetan Plateau (TP) located to the south" to "with the Tibetan Plateau (TP) located to the south".

**Response**: Thanks for your comments. According to your comments, we have modified the sentence as "The Tarim Basin (TB), is situated north of the Tibetan Plateau (TP), covers an area of  $5.3 \times 105$  km2, and contains the Taklimakan Desert (TD), which occupies the main part of the TB and is a major dust source in Asia." (**Lines 48-50**).

**3.** Line 62-65, A recent study also investigates the dust radiative impacts on vertical distribution of temperature and water vapor over Atlantic region, which is worthy to mention, Sun and Zhao (2020, Doi: 10.1029/2020JD033454).

**Response**: We have carefully read this paper. Sun and Zhao (2020) investigated the dust radiative impacts on vertical distribution of temperature and water vapor over Atlantic region. We have referenced it in our manuscript (**Lines 58-59**).

4. Line 72, I would suggest changing "present" to "presented"

**Response:** Thanks for your comments. We have modified the sentence as "Gu et al. (2006, 2016) and Law et al. (2006), based on numerical results, elucidated the impact of dust on air temperature in the upper layers" in the revised version (Lines 67-68).

5. Line 103, "reanalyzes" should be "reanalysis"?

**Response:** Thanks for your comments. We have changed "reanalyzes" to "reanalysis" (Line 97).

6. Line 129, As we know, it is more suitable to assume linear relationship between T and geometric altitude. When we use the altitude in hPa, I am not sure if the linear assumption between T and H is still robust or not. It might be okay when H (in hPa) varies within a small range.

**Response:** Thanks for your comments. In this study, the H (in hPa) varied within a small range (25 hPa), therefore, the linear assumption between T and H is still robust (Lines 123-125).

7. Line 130, "Where" should be "where".

**Response:** Thanks for your comments. We have corrected "Where" to "where" (Line 127).

8. Line 147-149, since the description here is similar as that at Lines 137-139, the authors might simply indicate that "The OMR value signal and magnitude have the same meanings as  $\Delta T$ ".

**Response:** Thanks for pointing out it. We have modified the sentence as "The signs and magnitudes of the OMR values have the same meanings as those of  $\Delta T$ " (Lines 148-149).

**9.** Line 157-158, the acronyms of the stations have already defined earlier, and no need to repeat.

**Response:** According to your comments, we have deleted full names of the stations here (Lines 156).

10. Line 158-159, this sentence might be also not necessary.Response: Thanks for pointing out it. We have deleted this sentence (Lines 157).

11. Line 177-178, I would suggest "radiative forcing"
Response: Thanks for pointing out it. We have changed "radiation" to "radiative" (Line 174).

12. Line 200, "were" is suggested as "are".

**Response:** Thanks for pointing out it. We have modified the sentence as "CALIPSO data revealed that vertically extended dust layers were widespread throughout the TB with peak lidar returns between 2.5 and 5.5 km above mean sea level due to strong convective activity during dusty as well as clear days (Cheng et al., 2020); this is consistent with other studies" (Lines 192-195).

13. Line 243, "present" should be "presents"

**Response:** Thanks for pointing out it. We have changed "present" to "presents" (Line 236).

**14.** Line 245-247, potential errors in ERA-5 reanalysis data are worthy to mention, while it is widely used and believed to be reliable.

**Response:** According to your comments, we have mentioned it in our manuscript in Lines 233-234.

15. Line 289, "suspend" should be "suspended"

**Response:** Thanks for your comments. We have corrected "suspend" to "suspended" (Line 279).

**16.** Line 295-299, the discussions here are great.

**Response:** Thanks so much for your kindness and your encouragement.

**17.** Line 341, "are" should be "is"

**Response:** Thanks for your comments. We have modified the sentence as "the average OMR values of the heating layer were 0.28 and 0.12 K in spring and summer, respectively" (Lines 333-335).

---

## Author Comment (AC2)

**Journal:** Atmospheric Chemistry and Physics

**Article title:** In-situ observation of warm atmospheric layer and the heat contribution of suspended dust over the Tarim Basin

**Manuscript number:** acp-2021-892

Dear Editors and Reviewers:

Thank you for your letter and for the reviewers' comments concerning our manuscript entitled "In-situ observation of warm atmospheric layer and the heat contribution of suspended dust over the Tarim Basin (acp-2021-892)". These comments are all valuable and very helpful for revising and improving our paper, as well as the important guiding significance to our researches. We have studied comments carefully and have made correction which we hope meet with approval. Revised portion are marked in the document named "manuscript_revised". The point-to-point responses to reviewer #2's comments are as follows.

**Reviewer #2:**

This work combined the radiosonde observations, reanalysis data as well as satellite images to investigate the possible impact of dust on meteorology. The authors found that there might be an anomalous warm atmospheric layer caused by suspended dust over the Tarim Basin, with a maximum heating effect of approximately +0.45 K and +0.25 K in spring and summer, respectively. The research topic is of interest. However, the descriptions on the utilized dataset ought to be detailed and also its applicability needs to be justified. I have the following questions and suggestions to the authors.

1. My main concern is that the authors claimed that "the ERA-5 data are generated from an ECMWF IFS spectral model and do not yet assimilate the impact of aerosols on meteorology", but "the MERRA-2 data include the impact of dust on meteorology". Actually, both the two data are reanalysis, which means that they have assimilated tremendous atmospheric observations, including temperature measurements. Here is the detailed information on the data assimilation system for

ERA-5 and MERRA2. The fact is that since 1997, ECMWF operations have applied 4D-var assimilation system.

https://www.ecmwf.int/en/elibrary/20196-ifs-documentation-cy47r3-part-ii-data-assimilation.

https://journals.ametsoc.org/view/journals/clim/30/14/jcli-d-16-0758.1.xml

These data assimilation systems do constrain the forecast by using surface observations, balloon data, aircraft reports, buoy observations, radar and satellite observations. Once the temperature and other meteorological fields are assimilated, the impact of aerosols on meteorology is certainly included in the reanalysis data. Investigations on relevant literature are highly suggested, based on which I also suggest the authors to reconsider the method or the datasets used in this work.

**Response:** Thanks for your comments. We agree with your opinion. Yes, as you mentioned above, the reanalysis data assimilation systems do constrain the forecast by using surface observations, balloon data, aircraft reports, buoy observations, radar and satellite observations. Generally, once the temperature and other meteorological fields are assimilated, the impact of aerosols on meteorology is certainly included in the reanalysis data.

In our study, we focused on the Tarim Basin (TB) region, which covers an area of $5.3 \times 10^5$ km$^2$ and contains the Taklimakan Desert (TD). And the TD is one of the major dust sources in Asia (Gong et al., 2003; Wang et al., 2005). However, at present, there are only 30 ground observation stations and 6 radiosonde stations in TB participated in global sharing (National Meteorological Information Center http://data.cma.cn/). To some degree, the scarce observational data could limit the quality of assimilation in both ERA-5 and MERRA-2 reanalysis data.

According to your suggestion, we have investigated the relevant literature. ERA-5 data indeed have assimilated multiple measurements through a four-dimensional variational data assimilation system in 12-hourly analysis cycles (Thepaut et al., 1996). However, ERA-5 data are generated from a spectral model (ECMWF Integrated

Forecast System) and have not considered the impact of aerosols on meteorology yet (Simmons, 2006). From the perspective of assimilation, reanalysis filed error includes model error and observation error (https://www.ecmwf.int/node/19997). Previous studies indicated that there are more than 100 dusty days per year in the TB (Zhou et al., 2020), meanwhile, these dust aerosols can suspend at attitude of 3-5 km for a long time, which have obvious positive radiation forcing, and the short-wave heating rate is greater than 6K/d (Huang et al., 2009). Therefore, if the effects of aerosol are not considered in the reanalysis model, the model error will be underestimated, which could somehow reflect the error in reanalysis field induced by dust aerosols.

Compared with ERA-5, MERRA-2 data include the assimilation of aerosol observations, thereby it provides a multidecadal reanalysis in which aerosol and meteorological observations are jointly assimilated within a global data assimilation system (Gelaro et al., 2017). More importantly, the aerosols and their interactions with weather and climate have been considered in MREEA-2 (Randles et al., 2017). Therefore, there is an obvious difference between ERA-5 and MERRA-2. Figure R1 shows a comparison of MERRA-2 data with radiosonde observations in the TB region. A good agreement is found between MERRA-2 data and radiosonde observations. Hence, we used the MERRA-2 reanalysis data to supplement the observation data.

[Figure]

Fig. R1. Comparison between atmospheric radiosonde temperature observations and MERRA-2 reanalysis data at altitudes of 500-300 hPa in spring and summer during 2016-2017.

The OMR method is proposed by Ding et al. (2021). This method is based on the assumption that the difference between observations and reanalysis models reflects the impact of un-resolved processes (Huang and Ding, 2021; Huang et al., 2018; Kalnay and Cai, 2003; Wang et al., 2013; Zhao et al., 2014). It means that these real biases, which result from missing physical or chemical processes in the model, have been misinterpreted as observational errors and discarded during the data assimilation procedure for the reanalysis data (Huang and Ding, 2021; Huang et al., 2018). As illustrated above, ERA-5 data were generated from a spectral model (ECMWF Integrated Forecast System) and has not considered the impact of aerosols on meteorology yet (Simmons, 2006). Therefore, investigation of the difference between observation (radiosonde observation and MERRA-2 reanalysis data) and ERA-5 reanalysis data can provide an opportunity to study the heating effect of dust aerosols, especially in a region with dust aerosol pollution (Huang and Ding, 2021; Ding et al., 2013). This method has been tested by previous work and proved to be well suited to identify the effects from aerosol impacts on the air temperature (Huang and Ding, 2021; Ding et al., 2013; Huang et al., 2018; Kalnay and Cai, 2003; Wang et al., 2013; Zhao et al., 2014). We have supplemented the reanalysis data and method in the revised version (**Lines 101-103; Lines 139-143**).

References:

Ding K., Huang, X., and Ding, A., et al., Aerosol-boundary-layer-monsoon interactions amplify semi-direct effect of biomass smoke on low cloud formation in Southeast Asia, Nat Commun., 12, 6416, 2021.

Ding, A. J., Fu, C.B., Yang, X.Q., Sun, J.N., Petäjä, T., Kerminen, V.M., Wang, T., Xie, Y., Herrmann, E., Zheng, L.F., Nie, W., Liu, Q., Wei, X.L., and Kulmala, M.: Intense atmospheric pollution modifies weather: A case of mixed biomass burning

with fossil fuel combustion pollution in eastern China, Atmos. Chem. Phys., 13, 10545-10554, 2013.

Gelaro, R., and Coauthors.: The Modern-Era Retrospective Analysis for Research and Applications, Version 2 (MERRA-2), J. Climate., 30, 5419-5454, https://doi.org/10.1175/JCLI-D-16-0758.1, 2017.

Gong, S.L., Zhang, X.Y., Zhao, T.L., Mckendry, I.G., Jaffe, D.A., and Lu, N.M.: Characterization of soil dust aerosol in China and its transport and distribution during 2001 ACE-Asia: 2. model simulation and validation, J. Geophy. Res., 108, D9, https://doi.org/ 10.1029/2002JD002633, 2003.

Huang, X. & Ding, A. J. Aerosol as a critical factor causing forecast biases of air temperature in global numerical weather prediction models, Sci. Bull., 66, 1971–1924, 2021.

Huang, X., Wang, Z.L., and Ding, A.J.: Impact of aerosol-PBL interaction on haze pollution: Multiyear observational evidences in North China, Geophys. Res. Lett., 45, 8596-8603, 2018.

Huang, J., Fu, Q., Su, J., Tang, Q., Minnis, P., Hu, Y., Yi, Y., and Zhao, Q.: Taklimakan dust aerosol radiative heating derived from CALIPSO observations using the Fu-Liou radiation model with CERES constraints, Atmos. Chem. Phys., 9, 4011-4021, https://doi.org/ 10.5194/acp-9-4011-2009, 2009.

Kalnay, E. & Cai, M. Impact of urbanization and land-use change on climate, Nature., 423, 528–531, 2003.

Randles, C.A., and Coauthors.: The MERRA-2 Aerosol Reanalysis, 1980 -- onward, Part I: System Description and Data Assimilation Evaluation, J. Climate., https://doi.org/ 10.1175/JCLI-D-16-0609.1, 2017.

Simmons, A.: ERA-Interim: New ECMWF reanalysis products from 1989 onwards, ECMWF newslett., 110, 25–36, 2006.

Thepaut, J. N., Courtier, P., Belaud, G., & Lemaitre, G. Dynamical structure functions in a four-dimensional variational assimilation: A case study. Quarterly Journal of the Royal Meteorological Society, 122(530), 535–561. https://doi.org/10.1002/qj.49712253012, 1996.

Wang, J., Yan, Z., Jones, P. D. & Xia, J. On "observation minus reanalysis" method: a view from multidecadal variability, J. Geophys. Res. Atmos., 118, 7450–7458, 2013.

Wang, S., Wang, J., Zhou, Z., and Shang, K.: Regional characteristics of three kinds of dust storm events in China, Atmos. Environ., 39, 509-520, https://doi.org/10.1016/j.atmosenv.2004.09.033, 2005.

Zhao, L., Lee, X., Smith, R. B. & Oleson, K. Strong contributions of local background climate to urban heat islands, Nature., 511, 216–219, 2014.

Zhou, C., Yang, F., Mamtimin, A., Huo, W., Liu, X., He, Q., Zhang, J., and Yang, X.: Wind erosion events at different wind speed levels in the Tarim Basin, Geomorphology., 107386, https://doi.org/10.1016/j.geomorph.2020.107386, 2020.

2. The northward extension of the Tibetan heat source proposed in 3.4 is somehow descriptive and hypothetical. I think in-depth data analysis and solid evidence should be provided while a scientific conclusion is drawn.

**Response:** Thank you for pointing it out. Yes, as you suggested, an in-depth data analysis and solid evidence are essential.

Gu et al. (2006, 2016) and Law et al. (2006) presented some numerical results to elucidate the dust's impact on air temperature at upper layers. As an observational evidence, Huang et al. (2009) found the significant radiative forcing and heating effect due to dusts over the part of the TB using CALIPSO data. However, there is no direct evidence about the heating of dust aerosols in the upper-layer atmosphere. In our study, the results show that there is a suspend dust layer over the TB in spring and summer (Figs 4 and 5), heating the atmospheric layer between 300-500 hPa with a maximum of approximately +0.45 K in spring and with a maximum of approximately +0.25 K in summer, which almost covers the whole basin (Figs 7c and 7d). Topographically, the TB is adjacent to the Tibetan Plateau (TP), which acts as an elevated heat source in spring and summer. The warm atmospheric layer over the TB seems a northward

extension of Tibet heat source. Therefore, the concept of the northward extension of Tibetan heat source is proposed. (**Lines 291-295**)

3. I also recommend language editing for improving the accuracy of language as well as overall readability.

**Response:** Thanks for your comments. We have proofed the language.

[Figure]

Minor issues

1. Line 75: What is "ground-based radiosonde observations"? Do you mean ground-based and radiosonde observations?

**Response:** Thanks for your comments. We have added "and" before "radiosonde observations" (**Line 70**).

2. Line 85-92 need to be rephrased.

**Response:** Thanks for your comments. We have rephrased it in the revised version as following:

We used radiosonde observations from six radiosonde stations situated in the TB for the spring and summer of 2016–2017 (Fig. 1), namely Kashi (KS), Akesu (AKS), Kuerle (KEL), Ruoqiang (RQ), Minfeng (MF), and Hetian (HT); from these data, we deduced the air-temperature profiles, which were measured twice per day (08:00 and 20:00 UTC+8). The observations were automatic and continuous, with 1 min temporal resolution, and the original data were processed into averages with 25 hPa interval. (**Lines 81-86**).

3. Line 104: correct reanalyzes to reanalysis

**Response:** Thanks for your comments. We have changed "reanalyzes" to "reanalysis" (**Line 97**).

4. Figure 3 is a little confusing and unclear since that it uses the size of markers to show the temperature difference. It might be more distinct while using gradient color.

**Response:** Thanks for your comments. We have modified Figure 3 as follows:

[Figure]

Fig. R3. Average $\Delta T$ between 500 and 300 hPa in (a) spring and (b) summer. Dots indicate the site locations, while different colors indicate different heating intensities

5.    The label in Figure 4 is too small to be clearly identified and needs to be improved. Figure 5 has the same problem.

**Response:** Thanks for your comments. We have modified Figures 4-5 as follows:

[Figure]

Fig. R4. The altitude–orbit cross-section of the 532-nm total attenuated backscattering intensity (left panels) and classified particles (right panels) on July (a1 and b1) 4, (a2 and b2) 5, (a3 and b3) 25, and (a4 and b4) 27, 2016, along the trajectory of the CALIPSO satellite over the Tibetan Plateau, as presented in Fig. 1. Gray shadings indicate the topography (**Page 32**).

[Figure]

Fig. R5. Frequencies of (a) dust events and (b) dust events including dust storms and cases of blowing and floating dust at the (a) Kashi (KS), (b) Akesu (AKS), (c) Kuerle (KEL), (d) Hetian (HT), (e) Minfeng (MF), and (f) Ruoqiang (RQ) stations in the spring and summer of 2016–2017; the blue dashed line represents the boundary between spring and summer. Seasonal distribution of the dust-top height in km over the Tarim Basin in (c) spring and (d) summer, inferred from CALIPSO (**Page 33**).

---

## Author Response (AR2)

**Journal:** Atmospheric Chemistry and Physics

**Article title:** In-situ observation of warm atmospheric layer and the heat contribution of suspended dust over the Tarim Basin

**Manuscript number:** acp-2021-892

Dear Editors and Reviewers:

Thank you for your letter and for the reviewers' comments concerning our manuscript entitled "In-situ observation of warm atmospheric layer and the heat contribution of suspended dust over the Tarim Basin (acp-2021-892)". These comments are all valuable and very helpful for revising and improving our paper, as well as the important guiding significance to our researches. We have studied comments carefully and have made correction which we hope meet with approval. The responses to reviewer #1's comments are as follows.

**Reviewer #1:**

My main comments have been well addressed, but the caption of Figure 5 needs to be double-checked. The labels of subplots and the locations of meteorological stations are quite confusing.

**Response:** Thanks for your comments. As you said that the labels of subplots and the locations of meteorological stations are quite confusing, we have revised it as follows: Taking KS as the start point, we arranged the stations clockwise.

[Figure]

Figure 5-R2. Frequencies of (a) dust events and (b) dust events including dust storms and cases of blowing and floating dust at the Kashi (KS), Akesu (AKS), Kuerle (KEL), Hetian (HT), Minfeng (MF), and Ruoqiang (RQ) stations in the spring and summer of 2016–2017; the blue dashed line represents the boundary between spring and summer. Seasonal distribution of the dust-top height in km over the Tarim Basin in (c) spring and (d) summer, inferred from CALIPSO.